# Role of the Chaperone Protein 14-3-3η in Regulation of the Infection Dynamics of the Influenza A (H1N1) Virus

**DOI:** 10.3390/v17101337

**Published:** 2025-09-30

**Authors:** Debarima Chatterjee, Partha Pratim Mondal, Anneshwa Bhattacharya, Alok Kumar Chakrabarti

**Affiliations:** ICMR-National Institute for Research in Bacterial Infections, Formerly (ICMR-National Institute of Cholera and Enteric Diseases), P33, CIT Road, Scheme XM, Beliaghata, Kolkata 700010, India; cdebarima@gmail.com (D.C.); partha348@gmail.com (P.P.M.); anneshwabhattacharya@gmail.com (A.B.)

**Keywords:** influenza virus A, H1N1, 14-3-3η, TRIM 32, polymerase basic 1 (PB1), antiviral

## Abstract

The 14-3-3 protein family, which includes the isoforms η, γ, ε, θ, β, and ζ, is essential for controlling a number of pathways linked to DNA and RNA viruses, including HIV, influenza A virus (IAV), measles virus, HRSV, and double-stranded DNA viruses. TRIM32, an E3 ubiquitin ligase, has been reported to target IAV’s PB1 polymerase for species-specific degradation via ubiquitination. Notably, 14-3-3η binds to phosphorylated TRIM32, preventing its autoubiquitylation and forming soluble but inactive cytoplasmic aggregates that regulate TRIM32 levels. However, the functional link between 14-3-3η, TRIM32, and PB1 during viral infection remains unclear. In this study, we establish a mechanistic connection between 14-3-3η–TRIM32 and TRIM32–PB1 interactions in IAV (H1N1) infection. We demonstrate that 14-3-3η directly interacts with PB1, influencing viral replication. Using transient knockdown models, we show that 14-3-3η deficiency alters influenza virus-induced cytotoxicity, cell death, immune responses, and reactive oxygen species (ROS) production. Additionally, we observe a significant reduction in the soluble TRIM32 levels in 14-3-3η-deficient cells, which leads to increased PB1 accumulation and thus suggests a critical regulatory role for 14-3-3η in PB1 stability. Our findings reveal a novel function of 14-3-3η in influenza virus infection, demonstrating its role in PB1 regulation via TRIM32 and its impact on innate immune activation. This study highlights 14-3-3η as a possible target for antiviral treatments against influenza and offers fresh insights into the host–virus relationship.

## 1. Introduction

Influenza is a transmissible viral disease that primarily affects the lower and upper respiratory tract of humans and other species [1]. Influenza A virus relies on close interaction with its host and uses host-cell proteins to promote its objectives. The IAV has developed several methods to block the innate immune response using diverse viral proteins after it infects the host body. IAV hijacks host cells’ molecular infrastructure to sustain its reproduction in the host body. This host–virus interaction is primarily perturbed by antiviral medicines. In the case of IAV, the PB1 and NS1 proteins play many critical functions in dealing with host immunity [2,3].

From yeast to mammals, 14-3-3 proteins are highly conserved across species and play a pivotal role in regulating diverse signalling pathways by binding to phosphorylated 14-3-3 recognition motifs on target proteins. Through these interactions, they critically influence intracellular signalling mechanisms involved in innate immunity and pathogen recognition. This interaction between viruses and 14-3-3 proteins may disrupt their typical distribution and interfere with their fundamental functions [4]. Distinct isoforms of 14-3-3 proteins, including η, γ, θ, β, ε, and ζ, participate in key cellular processes such as cell cycle regulation, apoptosis, signal transduction, and ubiquitination. Importantly, 14-3-3 proteins are implicated in the pathogenesis of several viral infections through their direct interactions with both DNA and RNA viruses, including influenza A virus, human respiratory syncytial virus, human immunodeficiency virus (HIV), measles virus, and double-stranded DNA viruses such as herpes simplex virus type I.

In the context of IAV, TRIM32, an E3 ligase, acts as an innate compulsive element that targets encounters with the PB1 polymerase after sensing for ubiquitination, which leads to degradation in a species-specific fashion [5]. Additionally, a potential autoregulatory mechanism may reduce the levels of soluble free TRIM32 within the cell, leading to the formation of cytoplasmic aggregates containing TRIM32. 14-3-3η proteins bind to phosphorylated TRIM32 and prevent the autoubiquitylation by protein-kinase-A-catalysed phosphorylation of TRIM32 at Ser65 [6]. 14-3-3 proteins also play a significant part in maintaining mitochondrial health in various ways, including protecting and regulating mitochondrial metabolism [6,7]. By controlling the retinoic acid-inducible gene I (RIG-I) translocator, a prime component in immunological signalling, these proteins help shield mitochondria from viral infections [7]. For example, during viral infection, 14-3-3ε interacts with influenza virus multifunctional NS1. By moving RIG-I to the mitochondria, this contact triggers the production of IFN, which has an antiviral impact [8,9,10]. While the NS3 protein of the dengue virus connects with 14-3-3ε, the NS3 protein of the Zika virus possesses a cellular 14-3-3-binding motif that can connect with 14-3-3η and 14-3-3ε [11,12].

During viral infections, 14-3-3η plays an important role in increasing the activation of MDA5-dependent antiviral innate immunity [13]. When MDA5 binds with the RNA viral nucleic acid, it activates downstream MAVS and functions as an intracellular viral sensor [14,15]. MAVS, a mitochondrial antiviral signalling protein, triggers caspase-9 and caspase-3 activation, which leads to apoptosis during viral infection. Caspase-9 is activated following the mitochondria’s release of cytochrome c. Caspase-3 and caspase-7, being the downstream executor caspases, are stimulated by active caspase-9 [14]. Caspase-3 is necessary for effective apoptosis and suppresses the generation of ROS. Influenza A can hijack caspases to aid in their propagation [14]. Overproduction of ROS can cause damage to the mitochondria and, as a result, apoptotic cell death. Acute lung damage and inflammation are caused by ROS-induced oxidative stress in influenza virus infections. On the other hand, viral interactions with mitochondrial membranes and other mitochondria-associated components increase the formation of reactive oxygen species (ROS). Virus-induced mitochondrial ROS (mtROS) stimulate viral reproduction by modifying host pathways and generating covalent changes in viral components [15]. Upregulation of 14-3-3η significantly reduced ROS and increased the levels of antioxidant enzymes in EPCs. 14-3-3η may prevent EPCs from decaying by reducing damage to the mitochondria [16]. There are several 14-3-3 protein isoforms in various species. The mitochondria are the nearly unique location of 14-3-3η in mammalian cells [17].

Our study reveals some unexplored involvement of 14-3-3η protein in cellular defensive mechanisms against influenza A viral infection progression, which affects innate immunity. Our experiments revealed that knockdown of 14-3-3η expression reduces IL-1a, IFN-β, and TNF-α expression in the case of influenza A virus infection but increases caspases’ activation, which enhances cellular ROS and mtROS production, as well as viral replication. In addition, we showed that upregulation of 14-3-3-η exclusively decreases mtROS production. On the other hand, PB1 and TRIM32 attachment could be decreased due to the knockdown of 14-3-3η expression. Our findings also indicate that 14-3-3η could foster PB1 degradation through TRIM32-mediated ubiquitination by involving viral PB1. Understanding these intrinsic mechanisms could stimulate the development of new anti-influenza A therapeutics.

## 2. Materials and Methods

### 2.1. Cells, Viruses, and Cloned Plasmids

Madin-Darby Canine Kidney (MDCK), Human Embryonic Kidney (HEK), and human lung adenocarcinoma (A549) cell lines were cultured and maintained in Dulbecco’s modified Eagle medium (DMEM) supplemented with 10% fetal bovine serum at 37 °C with 5% CO_2_ and humidity. Wild-type influenza *WSN/33*(H1N1) virus was used in this study. Overnight grown MDCK cell monolayers were infected with influenzza *WSN/33*(H1N1) virus and maintained at 37 °C with 5% CO_2_ and humidity for ~72 h until the appearance of considerable cytopathic effects. The harvested supernatant was clarified and viral quantification was performed using plaque assay and hemagglutination assay. The cDNA of full-length PB1, 14-3-3η, and TRIM32 was PCR amplified using a specific set of primers. Amplified PCR product purification was done using a PCR Purification Kit (QIAGEN) and PCR products were digested with restriction enzymes (*EcoR1*, *BamHI*, and *PstI*). Cloned plasmid constructs were made and checked, namely 14-3-3η–PCMV 6A (FLAG-tagged), PB1–GFP, TRIM32–PCMV 6A (FLAG-tagged), and TRIM32–DsRed (Figure 1). The specific sequences of primers used for the amplification of PB1, 14-3-3η, and TRIM32 cDNAs are mentioned in Table 1.

### 2.2. Reagents and Chemicals

Dulbecco’s modified Eagle’s medium (DMEM) was purchased from Invitrogen (Carlsbad, CA, USA). We purchased tissue culture flasks from Corning in the United States. 100 units/mL of streptomycin solution and fetal bovine serum (FBS), and 100 units/mL of penicillin, were purchased from Gibbco, (Oakland, CA, USA). We bought BV02, TPCK-treated trypsin, protease, and phosphatase inhibitor cocktail from Sigma-Aldrich (Burlington, MA, USA). TRIM32, MAVS, P16 primary antibodies, protein A/G agarose, and siRNA against 14-3-3η beads were bought from Santa Cruz Biotechnology (Santa Cruz, CA, USA). Primary antibodies of 14-3-3η and phospho-TRIM32 were purchased from (Thermo Fisher Scientific, Waltham, MA, USA). Primary antibodies for beta-actin, RIG1, MDA5, caspase 9, and caspase 3 were brought from (Cell Signalling Technology, Boston, MA, USA). ALEXA FLUOR 594 and 488 (secondary antibodies) were bought from Thermo Fisher Scientific, US. Influenza virus PB1 antibodies were purchased from Invitrogen. All secondary antibodies, including Lipofectamine 3000, were purchased from Invitrogen, United States. SYBR Green kt used n our study was purchased from the Applied Biosystems (Waltham, MA, USA). BioBharati Lifescience Pvt. Ltd. (Kolkata, WB, India) was the manufacturer of the Super Reverse Transcriptase MuLV Easy Kit used for the study. Promega Biotech supplied the CellTiter 96^®^ Non-Radioactive Cell Proliferation Assay (MTT) kit. MitoSOX Red and CM-H2DCFDA were from Invitrogen, USA. AgPath-ID™ One-Step RT-PCR reagents were purchased from Thermo Fisher Scientific, United States. All restriction enzymes were purchased from New England Biolabs (NEB), Ipswich, MA, United States.

### 2.3. Western Blotting

The protein turnover of the experimental cell lysates was examined by Western blotting. Following seeding on a 6-well plate, cells were either transfected, infected, or treated with the chemicals shown in the figures. After being treated with ice-cold PBS, cells were harvested, and cell lysis was performed using NP40 lysis buffer (150 mM NaCl, 50 mM Tris-Cl, 1% NP40, 1 mM PMSF). Protein estimation was done using the Bradford test and an equivalent quantity (~25 μg) of protein was loaded onto the wells of 10% SDS-PAGE for analysis. After that, the protein was electroblotted on a PVDF membrane for an hour at 4 °C at 100 V. After transferring the blot, the membrane was blocked for one hour at room temperature using 5% blocking buffer, which is made up of 5% BSA and 0.1% Tween-20 in TBS. The membrane was incubated overnight with specific primary antibody at 4 °C. Following incubation, the membrane was incubated with a secondary antibody for two hours at room temperature after being rinsed three times, for five to seven minutes each time, with TBS-T (TBS supplemented with 0.1% Tween 20). After three further TBS-T washes, the membrane’s protein bands were examined using the ECL chemiluminescence detection kit for HRP (Bio-Rad) as per the manufacturer’s instruction. A uniform amount of protein (30–50 μg) was loaded and assessed via SDS-PAGE, which was followed by the transfer of the protein to a PVDF membrane and immunoblotting utilizing specific primary antibodies at the concentration recommended by the manufacturer.

### 2.4. 14-3-3η Gene Knockdown by siRNA

HEK293 cells were seeded in 6-well plates at a density of ~2 × 10^5^ cells per well and allowed to adhere overnight. Cells were transfected with either 50 nM of 14-3-3η siRNA (h) (sc-43581, Santa Cruz Biotechnology) or universal control siRNA (sc-37007, Santa Cruz Biotechnology) using Lipofectamine 3000 (Invitrogen) according to the manufacturer’s protocol. Briefly, siRNA–lipid complexes were prepared in Opti-MEM (Gibco) and added to cells in antibiotic-free medium. After 6 h of incubation, the transfection medium was replaced with fresh complete DMEM supplemented with 10% FBS. Cells were then subjected to remedial treatment 48–72 h post-transfection.

### 2.5. 14-3-3 Protein-Protein Interaction Inhibition by Inhibitor BV02

We used BV02 side-by-side with siRNA to compare the effects of pan-inhibition of 14-3-3 versus isoform-specific knockdown of 14-3-3η. Prior to introducing BV02 into the experiment, we assessed its impact on HEK293 cells at different time points after infection to evaluate apoptotic induction. Additionally, we performed MTT assays to quantify cell viability under these conditions, ensuring accurate cytometry results. (Figure 2).

### 2.6. Immunoprecipitation

HEK293 cells were transfected with cloned plasmids as described in the figures by Lipofectamine 3000 reagent for the co-immunoprecipitation investigation. Twenty-four hours after transfection, the cells were lysed with NP40 lysis buffer on ice. From each batch, a 5% volume of the lysate was extracted for input. The total protein concentration in cell lysates was estimated using the Bradford protein assay (Bio-Rad), following the manufacturer’s instructions. Absorbance was measured at 595 nm using a microplate reader, and protein concentrations were determined by comparison with a standard curve generated from bovine serum albumin (BSA). The same amounts of lysates were used in a co-immunoprecipitation experiment. A total of 500 µg of protein from each lysate, with specific primary antibodies (1:50), was incubated using an end-to-end rotor that spun slowly for the entire night at 4 °C. On the second day, the immune complexes were centrifuged at 2000 RPM for precipitation. After that, protein A/G agarose beads were put on and gently rocked for four hours at 4 °C. NP-40 lysis buffer (half-diluted in PBS) was used to wash the immune complex-containing binding beads twice to eliminate any non-specific binding. Before the sample was loaded, the beads were boiled for 15 min with 4X SDS-PAGE gel loading buffer.

### 2.7. Reverse Transcription-qPCR

Total RNA was extracted from cells using the TRIzol reagent (Invitrogen) as directed by the manufacturer to determine the transcriptional expression of many genes in the investigation. Following total RNA extraction and quantification, complementary DNA (cDNA) was synthesized from 1 µg of RNA using the Super Reverse Transcriptase MuLV Easy Kit (Bio Bharati Life Science, India), according to the manufacturer’s protocol. The resulting cDNA was stored at −20 °C until further use. Then, SYBR Green (Applied Biosystems, MA, USA) was used to carry out the PCR reaction in the StepOnePlusTM Real-Time PCR System (Applied Biosystems) with primers specific to IL-1a, IFN-β, TNF-α, and 18s rRNA (Table 2); this cDNA was utilized to conduct the quantitative real-time polymerase chain reaction (qPCR). The following is the reaction protocol: 40 cycles of 95 °C for 15 s, 60 °C for 30 s, and 72 °C for 10 min after 50 °C for 2 min and 95 °C for 10 min. The relative expression of genes was computed and presented as a fold-change graph using the ΔΔCT method.

### 2.8. Viral RNA Extraction and RT-qPCR Analyses

Following the manufacturer’s instructions, viral RNA was extracted using the cellular supernatants and the QIAamp Viral RNA Mini Kit (Qiagen, Venlo, Netherlands). To prevent losses from freezing and thawing RNA extract and storage, all RNA samples were kept at −80 °C and analysed using RT-qPCR on the same day as RNA extraction. TaqMan-based RT-qPCR assay was used for IAV (H1N1) quantification. RT-qPCR analyses were performed using AgPath-ID™ One-Step RT-PCR Reagents. RT-qPCR mixture contained Supermix, viral-NA forward primer (5′- GAGCCCATATCGAACCCTAATG -3′), reverse primer (5′-GACCAAGCGACTGACTCAAA -3′), probe (5′-AGAGGGAACTTCACCAATAGGACAGC -3′), RT enzyme, and template RNA in StepOnePlus™ Real-Time PCR System (Applied Biosystems). The thermal cycling settings included denaturation and Taq polymerase activation at 95 °C for 10 min, RT at 50 °C for 30 min, 40 cycles of 95 °C for 15 s, and 65 °C for 1 min (data collection). For every instrument run, a standard curve was produced using the log10-linear regression of triplicate Cq data.

### 2.9. Immunofluorescence and Confocal Microscopy

Coverslips were used to seed HEK293 and A549 cells, which were then treated as shown in the figures. The cells were then fixed with 4% paraformaldehyde for 15 min and were permeabilized for 5 min with Triton X-100 (0.5%) in PBS. Blocking was done for 1 h at room temperature with 3% bovine serum albumin (BSA). Then, the cells were incubated with primary antibodies overnight in a moist chamber. Three subsequent PBS washes were conducted the next day, and the cells were left to react for one to two hours at room temperature with an anti-species-specific secondary antibody coupled with Invitrogen’s Alexa Fluor dye. After mounting the coverslips on slides, confocal microscopy was used to view the cells.

### 2.10. Analysis of Reactive Oxygen Species

The analysis of ROS was done by fluorescent microscopy and flow cytometry (BD FACSCalibur™ flow cytometer, Becton, Dickinson and Company, Franklin Lakes, NJ, USA). The indicators employed were CM-H2DCFDA for total cellular ROS and MitoSOX for mitochondrial ROS. HEK293 cells were treated with the appropriate drugs and transfected with the indicated constructs in 6-well plates. A cationic derivative of dihydroethidium, the MitoSOX reagent, is used to observe how mitochondrial superoxide enters living cells and targets mitochondria specifically. The red fluorescence-producing oxidation product of Mitosox can intercalate into mitochondrial DNA. A chemo-selective fluorescent naphthylimide peroxide probe was created to detect H_2_O_2_. For total cellular ROS, A 10 mM stock solution was created by dissolving 4.85 mg of CM-H2DCFDA in 1 mL of dimethyl sulfoxide (DMSO). Shortly before being added to the wells, the stock solution was diluted with preheated DMEM to provide a 10 μM working solution. A total of 500 μL of the DCFH-DA working solution was added to each well after the cells had been treated, as indicated in the figures. The wells were then incubated for 30 min at 37 °C. Then, the DCFH-DA working solution was removed and washed once with DMEM and 2× with 1×PBS. A total of 500 μL of 1× phosphate-buffered saline (PBS) was added to every well, and cells were harvested for FACS analysis.

### 2.11. Statistical Analysis

To determine the significance level of each group, data comparisons were done between groups using Student’s *t*-test for repeated measures. Every experiment has been carried out separately. The standard deviations (SDs) are shown in the figures as error bars. In the figures, the symbols *, **, ***, and **** denote *p* values less than 0.05, 0.01, 0.001, and 0.0001, respectively.

## 3. Results

### 3.1. Influenza Virus-Induced Cytotoxicity and Cell Death Are Increased in 14-3-3η-Deficient Cells

It is well known that influenza virus infection triggers the apoptotic signalling pathway, leading to cell death [18]. During virus infection, the members of the 14-3-3 family, specially the ‘eta’ isoform, play many crucial roles by augmenting and modulating many virus-sensing pathways and thus play a significant role in checkpoint regulation and signal transduction. To understand the role of the 14-3-3η protein in influenza virus infection dynamics, we performed a series of sequential experiments, as shown below.

We created 14-3-3η knockdown (KD) cells to ascertain how 14-3-3η affects influenza virus multiplication. Transient knockdown cells were produced by treating HEK293 and A549 cells with siRNA oligomers that target the 14-3-3η transcripts. Additionally, we employed BV02 as an inhibiter of 14-3-3 protein in our investigations to inquire into the inhibition of 14-3-3 PPI. After being infected by influenza A virus for 24 h, the infected cells exhibited the cytopathic effect (CPE), which manifests phenotypically as abnormal cellular structural morphology and the rounding-up and detaching of infected cells. Analysis of CPE development in virus-infected cells compared to the virus infection in the transduced cells revealed a considerably high infection rate in the case of 14-3-3η-deficient cells, which indicates role of 14-3-3η protein in inhibition of virus infection (Figure 1). Viral titres of the control and SiRNA-treated cells were determined by quantitative real-time RT-PCR, which indicated that the absence of 14-3-3η in infected cells induces virus infection. We found a significant increase in fold change in the viral titres in terms of the viral RNA content at different time points post-infection (h.p.i.) (Figure 2A). This result prompted us to further investigate whether there is any particular viral gene/protein that exhibits direct interplay with 14-3-3η’s activity in viral replication.

### 3.2. 14-3-3η Protein Interacts with the PB1 Protein of Influenza a Virus

Whether any of the influenza viral proteins induce the expression of 14-3-3η, we have measured the expression of 14-3-3η protein in cells transfected with the eight protein-expression plasmids of influenza A virus, as described in the Section 2. To investigate the role of 14-3-3η protein in relation to influenza viral genes, eight protein expression plasmids, each carrying one of the eight gene segments of the influenza virus, were individually transfected for 24 h into HEK293T cells using Lipofectamine 3000 (Invitrogen), following the manufacturer’s instructions. The lysates from the 24 h transfected cells were analysed by Western blot to obtain the expression of 14-3-3η protein. In parallel, lysates from control HEK293T cells were harvested at the same time point for comparison. Band intensities were quantified by densitometry using ImageLab 6.1 software, normalized to the loading control β-actin, and expressed as their fold change as compared to the control cells. The highest intensities of 14-3-3η protein bands were observed in the case of PB1 gene transfected cell sets as compared with the other seven genes, which indicates strong involvement of PB1 protein to induce 14-3-3 expression (Figure 2E,F). Further, we performed fluorescence microscopy to determine the PB1 expression in influenza virus-infected 14-3-3η-deficient cells. As shown in Figure 3A, both siRNA-treated and BV02 inhibitor-treated cells infected with influenza A virus showed significant upregulation of PB1 protein, which is directly related with high infectivity of the influenza virus in 14-3-3η-deficient cells, as shown by immunoblotting (Figure 2G–I). The infection efficiency, as measured by PB1 expression, was found to be highly upregulated at 12 h.p.i. as compared to 24 h.p.i. (Figure 3A). In fact, more fluorescence intensity could be visualized in the case of lower time points compared to higher ones. This apparent result made us more curious in detecting if PB1 is somehow directly involved or whether it interacts with 14-3-3η or not.

For further investigation, two consecutive experiments were performed: 14-3-3 protein was overexpressed in HEK293 cells by transfecting HEK293 cells with FLAG-tagged clones. These cells were infected with influenza A virus. Co-immunoprecipitation reaction between PB1 and 14-3-3η showed that PB1 directly interacts with 14-3-3 protein (Figure 3E). Overexpression of 14-3-3η in HEK293 cells followed by influenza A virus infection revealed a direct interaction between 14-3-3η and PB1, as demonstrated by co-immunoprecipitation (Figure 3E). This indicates that 14-3-3η may play a role in viral replication or protein complex formation.

We further performed confocal microscopic analysis to find interactions between the 14-3-3 and PB1 proteins, as mentioned in the Section 2. In brief, clones of the two proteins, 14-3-3η–FLAG and PB1–GFP, were co-transfected in HEk293 cells, while keeping the empty vectors as a control set. These experiments revealed the interactions between the two proteins, as shown in Figure 3B. The remaining cells from the slides were collected to prepare cell lysate to perform a Western blot analysis to validate the transfection efficiency. Figure 3C indicates high expression of both the proteins used in the above experiment. The intensity graph for the overlapped zone was also calculated and prepared using ImageJ 1.53t/Java 1.8.0_345 (64-bit) (Figure 3D). Confocal microscopy of HEK293 cells co-transfected with 14-3-3η–FLAG and PB1–GFP showed strong colocalization of the two proteins, whereas control cells expressing empty vectors did not (Figure 3B). Quantification of the overlapping fluorescence using ImageJ confirmed significant colocalization (Figure 3D), supporting the Co-IP results. Western blot analysis of the remaining cells validated the high expression of both proteins, confirming that the observed interactions were not due to variable transfection efficiency (Figure 3C). These results, together, suggest that 14-3-3η physically associates with PB1 and may be involved in influenza virus–protein interactions within host cells.

### 3.3. The Innate Immune Pathway, as Well as Cellular and Mitochondrial ROS Production, Is Modulated Due to IAV Infection in the Presence and Absence of 14-3-3η Protein

Our experiment showed that inhibition of 14-3-3 promotes viral replication and related cell death (Figure 2B). Viral-induced cell death has been documented proportionately related to cellular apoptosis. The primary agents responsible for carrying out the apoptotic response within the cell are caspases. Caspase 3 activation at the onset of apoptosis is a crucial event in the life-cycle of the influenza virus [19]. Our findings suggest that the caspase activity is reduced or delayed in cells lacking 14-3-3η. As shown in Figure 2B–D, after the treatment of 14-3-3η siRNA, caspase 3 expression was inhibited at 24 h post-infection as compared to control infection. However, in the later point of infection, when most of the cells have been apostatised, this effect of inhibition of caspase 3 by 14-3-3 protein is not significant. Reversely, 14-3-3η may have a role in promoting apoptosis of the infected cells, thereby preventing a longer replication period of the influenza A virus, which may be related to disease severity. Further studies are needed to prove this hypothesis. The graph plotted based on this observation signifies the fold change. Expression of NP was measured and a faster replication peak was achieved at 24 h.p.i. when the cells were treated with 14-3-3η siRNA, which indicates a quick replication in impaired 14-3-3η protein function. Therefore, 14-3-3 may have a role in controlling viral replication.

It has already been reported that PB1 mediates MAVS breakdown in an autophagosome-dependent way, which aids in virus replication and inhibits the antiviral innate immune response. Not only that, but 14-3-3 proteins themselves are essential for innate immunity [20]. They interact with retinoic acid-inducible gene I (RIG-I)-like receptors (RLRs), which are essential elements of the innate immune response to viral infections. In other words, they help modulate the immune response by regulating the activation and localization of key signalling proteins within the cell. We observed that, in the case of 14-3-3η siRNA-treated cells, major components of the innate immune system, such as MDA5–MAVS–caspase 9–caspase 3 stem, and their induced expression are delayed or modulated after 24 h of infection. To identify the impaired expression pattern of the following proteins, two sets of 14-3-3η siRNA-treated and untreated cells were infected with IAV, and lysates were collected at different time intervals. We selected the time points (8, 16, 24, and 30 h post-infection) after a literature study of the mean expression time of these proteins after any treatment on the cell. We conducted Western blot analysis of some proteins (RIG-1. MDA5, MAVS, caspase 9, caspase 3) which are related to innate immunity (Figure 4A). We also analysed them and plotted densitometric fold-change data (Figure 4B–F). The same observation is also found in the case of antiviral cytokine production. A suppressed number of antiviral cytokines like IL-1a, IFN-β, and especially TNF-α is observed in transduced cells after 24 h of infection (Figure 4G). This finding signifies that 14-3-3 regulates RIG-1. MDA5, MAVS, caspase 9, and caspase 3 undergo a series of interactions to delay apoptosis.

On the other hand, reactive oxygen species (ROS) production is accelerated by the involvement of viral interactions with mitochondrial membranes as well as other components associated with mitochondria. Viral components undergo covalent modifications, and host pathways are modulated by mitochondrial ROS (mtROS) induced by the virus, which promotes viral replication [21]. 14-3-3 proteins take part in mitochondrial health in a variety of ways. Among them, 14-3-3η proteins protect cells by reducing mitochondrial damage and mitophagy because 14-3-3η is almost exclusively localized to the mitochondria. Further, TRIM32 also regulates mitochondrial-mediated ROS levels and sensitizes oxidative stress-induced cell death, as reported. Our experiments depict that cellular ROS, as well as mtROS production, is increased not only in the case of IAV infection but also in the case of transfection with 14-3-3η siRNA followed by virus infection. In contrast, the application of a 14-3-3η clone for over-expression in HEK293 cells produces a lesser amount of mtROS (Figure 5).

### 3.4. Soluble TRIM32 Concentration Is Downregulated in 14-3-3η Knockdown Cells, Leading to Increased PB1 Accumulation

TRIM32, an innate IAV restriction factor, recognizes and directs the viral PB1 polymerase for degradation via ubiquitination. Cytoplasmic aggregates containing phosphorylated TRIM32 are normally stabilized by 14-3-3η, which prevents its auto-ubiquitylation. In our study, we observed that siRNA-mediated knockdown of 14-3-3η markedly reduced the soluble pool of TRIM32, which was reflected by fewer cytoplasmic TRIM32–14-3-3η complexes. This reduction was particularly evident in IAV-infected cells, where the soluble form of TRIM32 is essential to restrict viral propagation (Figure 6D). Consequently, the loss of soluble TRIM32 impaired PB1 restriction, leading to a higher accumulation of viral PB1 protein and enhanced viral infectivity. To confirm this, we performed co-immunoprecipitation of TRIM32 and PB1 in the presence or absence of 14-3-3η siRNA. Cells were co-transfected with TRIM32 and PB1 clones, and one set was pre-treated with 14-3-3η siRNA. Immunoprecipitation with PB1-specific antibody followed by TRIM32 immunoblotting revealed reduced TRIM32–PB1 binding in knockdown cells (Figure 6A). Fluorescence microscopy further confirmed these results, showing diminished colocalization of TRIM32 and PB1 upon 14-3-3η depletion (Figure 6B). Quantitative analysis of overlap using ImageJ corroborated the reduced interaction, as represented in the plotted graph (Figure 6C).

## 4. Discussion

14-3-3 proteins function as cytoplasmic regulatory molecules in all eukaryotic cells, primarily acting as scaffold proteins that modulate diverse cellular processes. Their significance in viral replication is well documented. For example, in influenza A virus, the ε isoform of 14-3-3 interacts with the NS1 protein to regulate innate immunity [9]. However, the role of 14-3-3η in influenza A virus infection has remained largely unexplored. Since 14-3-3η regulates soluble components of TRIM32—and TRIM32 itself is known to control PB1 polymerase activity in influenza A virus [5,6]—we hypothesized a potential interaction between 14-3-3η and viral proteins. Our initial observation of PB1’s interaction with 14-3-3η prompted us to investigate its functional significance during the influenza A virus life cycle.

We demonstrated that deficiency of 14-3-3η enhances cellular susceptibility to influenza virus infection, accelerating viral spread. Cells treated with 14-3-3η siRNA showed enhanced cytopathic effect ( CPE), which was corroborated by Western blot detection of viral proteins (PB1 and NP), confocal microscopy (Figure 1, Figure 2B and Figure 3A), and RT-PCR (Figure 2A). Consistent results were obtained upon pharmacological inhibition of 14-3-3 using BV02, which confirms the antiviral role of 14-3-3η. Cell viability assays (MTT) further revealed that 14-3-3η inhibition compromised cell survival in a time-dependent manner.

To identify which viral gene is responsible for inducing 14-3-3η, we transfected HEK293 cells with all eight protein expression plasmids of influenza A virus. Western blot analysis revealed that PB1 transfection produced the strongest 14-3-3η response. Confocal microscopy confirmed PB1 expression at both 12 h.p.i. and 24 h.p.i. (Figure 3A). Co-immunoprecipitation experiments further established a direct interaction between PB1 and 14-3-3η (Figure 3B).

The interplay between 14-3-3η and innate immune components was also investigated. MDA5 and MAVS, key members of the RIG-I-like receptor (RLR) pathway, mediate antiviral signalling upon viral RNA recognition [22,23,24]. MAVS, localized to mitochondria, activates caspase 9 and caspase 3, promoting apoptosis during viral infection [25,26]. However, PB1 has been reported to mediate MAVS degradation, thereby dampening host immunity [27]. Since 14-3-3η contributes to mitochondrial health [28], we analysed its role in modulating MDA5–MAVS–caspase signalling. Our data show that influenza virus infection enhances caspase 3 activity, which peaks around 30 h.p.i. Notably, 14-3-3η knockdown impaired cytokine induction (IL-1α, IFN-β, TNF-α), which is consistent with defective RLR signalling.

In addition, 14-3-3η knockdown increased cellular and mitochondrial ROS (mtROS) production, whereas 14-3-3η overexpression reduced mtROS accumulation. TRIM32 regulates mitochondrial ROS and restricts the influenza A virus by targeting PB1 for ubiquitination [29,30]. In our study, we examined how 14-3-3η influences TRIM32–PB1 interactions. Co-immunoprecipitation and fluorescence microscopy revealed that 14-3-3η knockdown decreased TRIM32–PB1 binding, likely due to reduced soluble TRIM32 levels (Figure 6A–D).

Taken together, our results indicate that 14-3-3η regulates influenza A virus replication by stabilizing TRIM32–PB1 interactions, maintaining mitochondrial integrity, and supporting innate immune signalling. Importantly, 14-3-3η knockdown facilitates viral replication by promoting MAVS degradation, ROS accumulation, and impaired cytokine production. Given its multifaceted antiviral role, therapeutic approaches aimed at enhancing 14-3-3η expression or activity may represent a novel strategy against the influenza virus. However, further in vivo validation and exploration of pharmacological modulators of 14-3-3η are required before clinical translation.

## 5. Conclusions

Interactions of 14-3-3η with the host protein TRIM32 and the viral PB1 protein have been studied in the context of influenza virus infection. Collectively, our results demonstrated that 14-3-3η plays a crucial role in influenza virus infection dynamics by regulating PB1 protein degradation through TRIM32. During influenza A virus infection, 14-3-3η plays a critical role in modulating cellular innate immunity. This study displayed an immunological response triggered by influenza virus infection through the MDA5–MAVS–Caspase 9–Caspase 3 axis of the innate immune pathway. The role of 14-3-3η in protecting mitochondrial health by decreasing mitochondrial ROS production has been evaluated in this study. Very importantly, it could aid in the discovery of new therapeutic targets for the influenza virus. Our research ultimately contributes to a better knowledge of the pathophysiology of influenza viruses and the host’s defences against influenza virus infection.

## Data Availability

The datasets used and/or analyzed during the current study are available from the corresponding author on reasonable request.

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
