# Peer review of "Role of the Chaperone Protein 14-3-3η in Regulation of the Infection Dynamics of the Influenza A (H1N1) Virus"

_viruses, 2025, doi:10.3390/v17101337_

Round 1

Reviewer 1 Report

Comments and Suggestions for Authors

This article presents a possible influential role for the eukaryotic protein 14-3-3η in IAV infection. The authors demonstrated the role of this protein over cytotoxicity, cell death, and the immune response of virus in culture cells.

Although the work is very thorough, the main recommendations for improvement are based on the organisation of the information and improving the clarity with which the results are presented:

Introduction section: I would recommend improving the wording for greater fluency and adding references. For example, up to line 14 of the second paragraph, a wealth of information is provided about the 14-3-3 protein family without any associated references.

Materials and methods: The number of experiments performed in each case should be indicated in order to assess the suitability of the statistical test used. This is implied in the discussion section, but it is not the appropriate place to indicate such information.

The figures should be ordered according to the order in which they appear in the text. For example, in the first text of the results, reference is made to Figures 1A and 2A, and Figure 1B is not explained until the end of the second paragraph of results, after Figure 2 and after reference has been made to Figures 1D and 1E. This is an example of disorder that is repeated throughout the paper.

In the discussion section, the authors do not do a real discussion of the results presented in comparison with other findings. For the most part, they repeat what was done in the study and the results obtained, but there is a general lack of references to other studies with similar or conflicting results and a proper discussion of them.

Author Response

We are grateful to the reviewer for investing valuable time in reviewing our manuscripts. Your insightful comments greatly helped us in revising the manuscript, which we believe have enhanced the quality of the manuscript. We have carefully gone through all the comments and addressed each of the comments in a sequential manner.

Reviewer 1

Comments and Suggestions for Authors:

This article presents a possible influential role for the eukaryotic protein 14-3-3η in IAV infection. The authors demonstrated the role of this protein over cytotoxicity, cell death, and the immune response of virus in culture cells.

Comment: Although the work is very thorough, the main recommendations for improvement are based on the organisation of the information and improving the clarity with which the results are presented:

Response: Thank you so much for your kind input. In accordance with the suggestions and remarks from the reviewers, we have made revisions to the manuscript. The updated manuscript has been structured according to the insights derived from the results, presented in a significantly clearer manner.

Comment: Introduction section: I would recommend improving the wording for greater fluency and adding references. For example, up to line 14 of the second paragraph, a wealth of information is provided about the 14-3-3 protein family without any associated references.

Response: Thank you so much for the valuable comment. As per your suggestion we have incorporated appropriate references (Reference No. 4 and 5) about 14-3-3 protein family which has been reflected in the revised manuscript.

Comment: Materials and methods: The number of experiments performed in each case should be indicated in order to assess the suitability of the statistical test used. This is implied in the discussion section, but it is not the appropriate place to indicate such information.

Response: Thanks for the insightful comment. As per the suggestion we have mentioned the desired information under the subheading Statistical analysis in the material and method section.  We have revised the discussion section as per your recommendations and those of other reviewers; consequently, the information regarding statistical tests is no longer included in the Discussion section.

Comment: The figures should be ordered according to the order in which they appear in the text. For example, in the first text of the results, reference is made to Figures 1A and 2A, and Figure 1B is not explained until the end of the second paragraph of results, after Figure 2 and after reference has been made to Figures 1D and 1E. This is an example of disorder that is repeated throughout the paper.

Response: We sincerely appreciate your comment. In accordance with your suggestion, we have reorganized the figures into the correct sequences to better align with the relevant text in the revised manuscript. For instance, we have condensed Figure 1 and transferred a segment of its content to Figure 2 (Fig 2 E-I) in the revised manuscript to ensure consistency between the text and figure contents. Although the figure numbers remain the same, the content within Figures 1 and 2 has been rearranged in the revised manuscript. We have thoroughly reviewed the entire manuscript and believe that the text descriptions and figures are now properly aligned.

Comment: In the discussion section, the authors do not do a real discussion of the results presented in comparison with other findings. For the most part, they repeat what was done in the study and the results obtained, but there is a general lack of references to other studies with similar or conflicting results and a proper discussion of them.

Response: Thanks for the valuable suggestion which has enhanced both the quality of the manuscript and the presentation of the work. In accordance with your recommendation, we have rewritten the discussion to better reflect the theme of the work and have included appropriate references in the relevant areas of discussion.

Reviewer 2 Report

Comments and Suggestions for Authors

The manuscript “Role of the chaperone protein 14-3-3η in regulation of the Influenza A (H1N1) Virus Infection dynamics” describes a mechanistic connection between key viral and host factors involved in influenza infection, namely 14-3-3η, TRIM32 and PB1. 14-3-3η has been shown to directly interact with viral polymerase subunit PB1, its deficiency altering influenza virus-induced cytotoxicity, cell death, immune responses and reactive oxygen species production. In general, the results obtained provide novel information about the host-virus relationship as well as determine 14-3-3η as a possible target for antiviral treatments against influenza.

Prior to publication, several issues should be addressed. Serious correction of style and grammar should be done.

Footnotes for figures must be provided, as no such information can be found in the submission.

Page 3, third paragraph. Authors should provide the specific sequences of primers used for and protocol of amplification of PB1, 14-3-3η, and TRIM32 cDNAs.

Page 3, third paragraph. Change to read “… for Ì´72 hours, and the virus was quantified…”

Page 4, second paragraph. Authors should provide the specific sequences of siRNA and protocol of the procedure used for 14-3-3η siRNA knockdown.

Page 4, section “Immunoprecipitation”. “The quantity of protein in the cell lysate was estimated.”. Please provide the protocol of or reference to the procedure of protein quantification.

“A 500 μg concentration of the entire lysate…” – does this mean 500 μg/mL? If so, please correct.

Page 4, last paragraph. Change to read “… (cDNA) was produced by using…”

Figure 1A. The difference between three parts of the figure can be hardly understood. The issues reader’s attention to be paid should be described and highlighted.

Page 8. “To understand the involvement of 14-3-3η protein with viral genes, we used 8 different reverse genetics plasmids carrying 8 gene segments of the influenza virus,…” Please re-phrase. This is hardly to understand.

Page 8. “Band intensities were analysed and normalized them with the loading control beta-actin…” Please correct the grammar, here and further.

Page 10. “…two consecutive experiments were performed: 14-3-3 protein was overexpressed in HEK293 cells by transfecting HEK293 cells with Flag-tagged clone…”. Why authors consider this experiment as two?

Page 10, second paragraph. The results and their significance should be described. Now there is just a describing of the methods used and what has been done, without providing of the results.

Page 10, third paragraph. “Results of our experiment is indicative…” Please re-phrase.

Author Response

Point to point response to the reviewers

We are grateful to the reviewer for investing valuable time in reviewing our manuscripts. Your insightful comments greatly helped us in revising the manuscript, which we believe have enhanced the quality of the manuscript. We have carefully gone through all the comments and addressed each of the comments in a sequential manner.

 Reviewer 3

Comments and Suggestions for Authors

Comment: Footnotes for figures must be provided, as no such information can be found in the submission.

Response: Thank you for the concern. We have forwarded the legends of figures as a separate file via e mail after submission which you may not have received. We have included the detailed explanations of the figures as legends in the revised manuscript.

Comment: Page 3, third paragraph. Authors should provide the specific sequences of primers used for and protocol of amplification of PB1, 14-3-3η, and TRIM32 cDNAs.

Response: Thank you so much for the input. We have provided the specific primer sequences used in this study and the protocol of amplifications. Kindly follow Table: 1 of Page 3.

Comment: Page 3, third paragraph. Change to read “… for Ì´72 hours, and the virus was quantified…”

Response: We appreciate your input. The specified line has been revised. Please refer to page No. 3, Line-97-99.

Comment: Page 4, second paragraph. Authors should provide the specific sequences of siRNA and protocol of the procedure used for 14-3-3η siRNA knockdown.

Response: Thank you a lot for the concern. It was our mistake to not to provide the necessary details. We have now included the siRNA information. Please refer to Page 6, Line 150. However, the specific sequence could not be retrieved as it was acquired from Santa Cruz Biotech, which does not disclose it. Therefore, we have provided the name and catalog number associated with the product utilized in our experiment.

Comment: Page 4, section “Immunoprecipitation”. “The quantity of protein in the cell lysate was estimated.”. Please provide the protocol of or reference to the procedure of protein quantification.

Response: Thank you so much for the concern. The segment has been revised. Please refer to the yellow highlighted section on Page 6, Lines 163-167 of the updated submission.

Comment: “A 500 μg concentration of the entire lysate…” – does this mean 500 μg/mL? If so, please correct.

Response: Thank you for the response. It actually does mean “A total of 500 µg of protein from each lysate...” This change has been reflected in line 168 of the revised manuscript.

Comment: Page 4, last paragraph. Change to read “… (cDNA) was produced by using…”

Response: We sincerely appreciate your feedback. The specified line has been revised. Please refer to Page No. 6, Lines 180-183.183.

Comment: Figure 1A. The difference between three parts of the figure can be hardly understood. The issues reader’s attention to be paid should be described and highlighted.

Response: We are grateful for your valuable feedback. Figures 1 and 2 have been rearranged for greater accuracy.

We have condensed Figure 1 for improved clarity, and a portion of Figure 1 (Fig 1E-I) has been incorporated into Figure 2. This aligns well with the text and enhances the overall significance of the manuscript.

Comment: Page 8. “To understand the involvement of 14-3-3η protein with viral genes, we used 8 different reverse genetics plasmids carrying 8 gene segments of the influenza virus,” Please re-phrase. This is hardly to understand.

Response: Thank you a lot for the input. We have reframed the Paragraph portion. Kindly check Page 9, Line-277-282.

Comment: Page 8. “Band intensities were analysed and normalized them with the loading control beta-actin…” Please correct the grammar, here and further.

Response: Thank you so much for the input. We have re written the portion. (Lines 283-285, page 9 in the revised manuscript)

Comment: Page 10. “…two consecutive experiments were performed: 14-3-3 protein was overexpressed in HEK293 cells by transfecting HEK293 cells with Flag-tagged clone…”. Why authors consider this experiment as two?

Response: Thank you for the valuable concern. We described it as “two consecutive experiments” because, conceptually and technically, they are investigating the 14-3-3η–PB1 interaction using two different experimental approaches, even if both involve HEK293 cells:

  1. Co-immunoprecipitation (Co-IP) experiment:
  • Purpose: To determine whether 14-3-3η physically binds to PB1 at the biochemical level.
  • Output: Direct evidence of protein–protein interaction through Western blot of immunoprecipitated complexes (Fig. 3E).
  1. Confocal microscopy experiment:
  • Purpose: To visualize the subcellular localization and colocalization of 14-3-3η and PB1 in intact cells.
  • Output: Microscopy-based spatial evidence of interaction and overlap between the two proteins (Fig. 3B–D).

Even though both experiments investigate the same biological question (interaction between 14-3-3η and PB1), the techniques, readouts, and information provided are different: one gives biochemical evidence (Co-IP), the other gives spatial/visual evidence (confocal microscopy). In short, we consider them two experiments because they complement each other with different approaches and types of evidence, not because they are two separate manipulations of the cells.

Comment: Page 10, second paragraph. The results and their significance should be described. Now there is just a describing of the methods used and what has been done, without providing of the results.

Response: Thank you so much for the concern. We have Re-written the concerned portions according your suggestions. The result portion and the discussion now showing a clearer view. Kindly check the paragraphs which are highlighted yellow (Page No.11, Line No. 372-389 of the revised manuscript).

Comment: Page 10, third paragraph. “Results of our experiment is indicative…” Please re-phrase.

Response: Thank you for the input. We have re-phrased this line as per your suggestion. (Page 10, Lines No 328-329 of the revised manuscript)

Round 2

Reviewer 2 Report

Comments and Suggestions for Authors

Authors have addressed all issues previously expresssed. The manuscript can be published in its current form.